# Antimicrobial Stewardship Techniques for Critically Ill Patients with Pneumonia

**DOI:** 10.3390/antibiotics12020295

**Published:** 2023-02-01

**Authors:** Jenna Adams, Kaitlin Ferguson, RaeAnn Hirschy, Erica Konopka, Jordan Meckel, Grace Benanti, Shannon Kuhrau, Fritzie Albarillo, Kevin Chang, Maressa Santarossa, Julia Sapozhnikov, Brian Hoff, Megan A Rech

**Affiliations:** 1Department of Pharmacy, Loyola University Medical Center, 2160 S First Ave, Maywood, IL 60153, USA; 2Department of Medicine, Division of Infectious Diseases, Loyola University Medical Center, 2160 S First Ave, Maywood, IL 60153, USA; 3Department of Emergency Medicine, Stritch School of Medicine, Loyola University Chicago, 2160 S First Ave, Maywood, IL 60153, USA

**Keywords:** pneumonia, antimicrobial stewardship, critically ill, methicillin-resistant *Staphylococcus aureus*, *Pseudomonas aeruginosa*, procalcitonin

## Abstract

Pneumonia is common in the intensive care unit (ICU), infecting 27% of all critically ill patients. Given the high prevalence of this disease state in the ICU, optimizing antimicrobial therapy while minimizing toxicities is of utmost importance. Inappropriate antimicrobial use can increase the risk of antimicrobial resistance, *Clostridiodes difficile* infection, allergic reaction, and other complications from antimicrobial use (e.g., QTc prolongation, thrombocytopenia). This review article aims to discuss methods to optimize antimicrobial treatment in patients with pneumonia, including the following: procalcitonin use, utilization of methicillin-resistant *Staphylococcus aureus* nares testing to determine need for vancomycin therapy, utilization of the Biofire^®^ FilmArray^®^ pneumonia polymerase chain reaction (PCR), and microbiology reporting techniques.

## 1. Introduction

Pneumonia is a common etiology and complication of critical illness, affecting around 27% of all critically ill patients according to an article published in 2006 [1]. Pneumonia is classified via four different categories: non-severe community-acquired pneumonia (CAP), severe CAP, hospital-acquired pneumonia (HAP), and ventilator-associated pneumonia (VAP). Each of these categories has varying recommendations for antimicrobial therapy. For HAP and VAP, empiric broad-spectrum antimicrobials are recommended, while for CAP, broad-spectrum antimicrobials are recommended for certain patients that experience certain risk factors for multi-drug-resistant organisms [2,3]. This leads to many patients being treated empirically with antimicrobials active against *Pseudomonas sp.* and frequently methicillin-resistant *Staphylococcus aureus* (MRSA).

In a prospective review, 526 nasal swabs and 506 pharyngeal swabs were assessed for the effects of hospitalization on endogenous bacterial flora in the nasal cavity, as well as pharyngeal space. After 7–14 days of hospital admission, nasal and pharyngeal flora significantly changed, leading to the isolation of more Gram-negative organisms. Gram-negative isolation increased from 6% to 18.8%, while MRSA isolation increased from 2.1% to 7.5% [4]. The most common causative organism in patients with pneumonia that have been hospitalized for less than 5 days is *Streptococcus pneumoniae*, while the most common organisms associated with hospital-onset pneumonia (HAP and VAP) included MRSA (10–20%), *Pseudomonas aeruginosa* (PSAR) (10–20%), enteric Gram-negative rods (20–40%), and *Acinetobacter baumannii* (5–10%) [2,5]. These data support guideline recommendations to provide initial antimicrobial coverage for MRSA and PSAR in patients with HAP and VAP [2].

Given the high prevalence of this disease state in the intensive care unit (ICU), optimizing antimicrobial therapy while minimizing toxicity is of utmost importance. One negative consequence of antimicrobial use is the development of resistant organisms. The Centers for Disease Control and Prevention (CDC) describes antimicrobial resistance as an “urgent global public health threat, killing at least 1.27 million people worldwide and associated with nearly 5 million deaths in 2019”. The use of second- to third-line antimicrobials that are necessary for multi-drug-resistant isolates can result in serious health complications including organ failure and prolonged care [6]. A retrospective cohort study including 1816 patients, who received at least one dose of cefepime, meropenem, or piperacillin-tazobactam, found that as cumulative days of anti-pseudomonal exposure increased, the risk of resistance to anti-pseudomonal beta-lactams also increased. For patients that were exposed to one to three days of anti-pseudomonal beta-lactams, resistance developed in 2.1% (HR = 1; reference sample). In patients receiving seven to nine days of therapy, the risk of developing resistance increased to 7.9% (HR 1.01 [95% confidence interval (CI)s 0.93–1.1]). Resistance was further increased in patients who received 19–21 days of anti-pseudomonal beta-lactam therapy to 11.5% (HR = 7.05 [95% CI 6.19–8.02]) [7]. A similar retrospective cohort study was undertaken to evaluate incidence of subsequent carbapenem-resistance *Acinetobacter baumanii* infections (CRAB) following administration of third-generation cephalosporin or carbapenems. The use of third-generation cephalosporins (*p* < 0.001; regression co-efficient 0.006 [95% CI 0.004–0.009]) and carbapenems was correlated with development of CRAB infections (*p* = 0.014; regression co-efficient 0.002 [95% CI 0.001–0.004]) [8]. The findings from these two studies highlight the importance of antimicrobial stewardship to prevent the development of multi-drug-resistant organisms.

Aside from the risk of antimicrobial resistance, adverse effects from antimicrobials can include *Clostridiodes difficile* infection, antimicrobial allergic reaction, as well as other complications such as QTc prolongation, thrombocytopenia, and renal impairment. As such, correctly identifying the type of pneumonia (HAP, VAP, non-severe CAP, severe CAP) and the patient’s risk for multi-drug-resistant (MDR) organisms is the first step of appropriate antimicrobial prescribing. This review article aims to discuss diagnostic tools to optimize antimicrobial treatment in patients with pneumonia, including the following: procalcitonin (PCT), MRSA nares testing to determine need for vancomycin therapy, the Biofire FilmArray^®^ pneumonia polymerase chain reaction (PCR), and microbiology reporting techniques.

## 2. Utilization of Procalcitonin

### 2.1. Clinical Relevance

PCT, a 116-amino-acid peptide, is a precursor to calcitonin and normally secreted from the thyroid gland. In the setting of bacterial infection, PCT levels rise rapidly due to production in extra-thyroid tissue [9]. The rise in PCT stems from a wide variety of causes, all of which result in a systemic inflammatory response, including severe trauma or burns, pulmonary aspiration, bowel ischemia, and surgical procedures [10]. Since its discovery in the 1990s, PCT has been a potential tool for antimicrobial stewardship for various types of pneumonia, with mixed results in the published data [11,12,13,14,15]. Elevated PCT is perpetuated by the presence of other inflammatory cytokines (e.g., interleukin (IL)-1β, tumor necrosis factor (TNF)-α, and IL-6). Studies of PCT have defined “elevated” in different ways, with the most common being PCT > 0.5 ng/mL. Consensus cutoffs in the literature are generally as follows: PCT < 0.1 ng/mL, antibiotics strongly discouraged; PCT < 0.25 ng/mL, antibiotics discouraged; PCT > 0.25 ng/mL, antibiotics encouraged; PCT > 0.5 ng/mL, antibiotics recommended [16].

The 2016 Infectious Disease Society of America (IDSA) HAP and VAP guidelines and the 2021 Surviving Sepsis guidelines both recommend against using procalcitonin as a diagnostic tool to prompt antibiotic initiation (strong and weak recommendations, respectively) [2,17]. Studies since these guidelines were published also reinforce these recommendations. In a retrospective cohort, 332 patients with bacteremia in the first 24 h of admission had a procalcitonin drawn within 48 h. Using a common threshold of 0.5 ng/mL, it was concluded that procalcitonin had a 62% sensitivity for bacteremia, which is lower than a previously reported 76% using the same threshold [18,19]. Interestingly, those that had a PCT < 0.5 were less likely to exhibit leukocytosis, require mechanical ventilation or vasopressor support, have a respiratory rate of 20 breaths per minute or greater, or require ICU care. The PCT < 0.5 ng/mL group also had significantly more infections of the skin or bone, which are typically localized infections without systemic symptoms where PCT utility is limited [20]. Despite recommendations against routine use as a diagnostic tool, procalcitonin may be considered in certain clinical scenarios to support objective antibiotic de-escalation as described further in the sections that follow.

### 2.2. Impact on Antimicrobial Stewardship

#### 2.2.1. Timing

Time of assessment of PCT is a critical component of interpretation. An in vivo study simulating an infectious process using *Escherichia coli* endotoxin injected into healthy volunteers found serum PCT levels began to rise four hours after injection, while symptoms of systemic infection began after one to three hours [21]. Several studies suggest PCT elevation may not occur until six hours after contact with bacteria; however, these studies did not routinely sample PCT prior to six hours from the initial baseline level [22,23]. In a real-world study of healthy volunteers injected with *E. coli* endotoxin, serum levels were detectably elevated three hours after insult and had peaked after 24 h [24]. Based on data from these studies, the half-life of PCT is estimated to be around 24–48 h. Since PCT levels rise soon after symptoms typically develop, the notion of a negative result being drawn too early (e.g., falsely negative early in disease) is unlikely in a symptomatic patient. Drawing a PCT prior to systemic symptoms, however, will likely result in the patient’s normal range PCT and thus is not recommended. If the suspected infection is localized and lacking in systemic signs and symptoms, PCT may not change significantly from baseline regardless of timing.

#### 2.2.2. False Elevation

The utility of PCT as an antibiotic stewardship tool lies in its ability to discern the likelihood of a systemic infection being present or absent. PCT lacks specificity in the setting of other systemic inflammatory processes [25]. Patients with burns, trauma, major surgery, invasive fungal infections, and other causes of systemic inflammatory responses with a PCT of greater than 0.5 ng/mL should be more critically examined when making antibiotic decisions. A prospective study of 751 patients evaluated the prognostic capacity of PCT for post-operative complications after cardiac surgery. In multivariate analysis, a PCT greater than 2.95 ng/mL had a greater than 60-fold increase in rates of complications both infectious and non-infectious in nature [26]. Thus, PCT may be useful in predicting post-operative complications, although specific cutoff levels for post-operative infection have not been determined.

Renal dysfunction is a common comorbidity in critically ill patients with published rates that exceed 50% [27,28,29]. In a meta-analysis evaluating PCT in patients with varying degrees of chronic kidney disease (CKD), PCT-guided care reduced antibiotic exposure with no effect on mortality [30]. However, the pooled baseline PCTs ranged drastically from <0.1 to >2. Of the CKD stage 5 patients, nearly half of the control and experimental groups had baseline PCT > 2 ng/mL, whereas the CKD stage 1 group was between 26.8% and 29.4%. Higher-than-normal PCTs have been noted in non-infected patients with CKD and should be excluded from PCT-driven antibiotic stewardship protocols that use hard cutoff values to guide decisions on antibiotic initiation [31,32]. Trending PCT in CKD patients for antibiotic de-escalation purposes may still provide clinical utility, as PCT metabolism/clearance is only loosely correlated with renal function [33].

#### 2.2.3. De-Escalation Tool

While PCT lacks the sensitivity needed for diagnostic purposes, the use of PCT has been postulated as an antibiotic de-escalation or cessation tool in the ICU. A recent meta-analysis specifically examined the effects of PCT-guided antibiotic initiation and cessation approaches separately in addition to mixed approaches in critically ill patients in an attempt to reduce confounding results and heterogeneity. Nine of the fifteen studies included focused on PCT-guided antibiotic cessation and concluded that PCT-guided antibiotic cessation protocols were associated with decreased mortality. Study protocols varied widely and included both exact value cutoffs between 0.1 mcg/L to 0.5 mcg/L and changes in procalcitonin from initial level, which ranged from a 65–75% decrease to >90% without indicating which type of threshold was utilized for each patient [34]. It is notable that there are innumerable reasons for PCTs to be elevated in settings other than bacterial infections, making the utility of exact value cutoff cessation protocols difficult to generalize in critically ill populations. Further studies comparing the NPV of published protocols of PCT downtrends from baseline in critically ill patients will be helpful in identifying a generalizable strategy. The most conservative measures used in the studies included by Lam et al. utilized a de-escalation threshold of 80–90% PCT reduction to discontinue antibiotics in lieu of newer, isolated studies; this is a reasonable approach to implement as an antibiotic stewardship measure in the ICU.

### 2.3. Conclusions

The United States Food and Drug Administration approved the use of PCT in the management of lower respiratory tract infections and antibiotic de-escalation in sepsis. However, published data are conflicting on the exact role of PCT as a tool for infectious diseases diagnostics and antimicrobial decision making. The sensitivity and NPV for PCT in these disease states are variable. Generally, an elevated PCT should not be used alone to distinguish the presence of bacterial infection in lower respiratory tract infections, and the results must be interpreted cautiously within the context of other diagnostics, medications, and comorbidities. With regards to timing, in patients with lower respiratory tract infections, the PCT should be drawn early upon presentation and trended sequentially for antibiotic de-escalation. Overall, PCT has limited utility in antimicrobial initiation, and clinicians should be cautious with interpretation of results to guide diagnosis and initial antibiotic prescribing. However, PCT may provide a benefit for antimicrobial stewardship in the ICU as a de-escalation tool.

## 3. Methicillin-Resistant *Staphylococcus aureus* Nasal Polymerase Chain Reaction Utilization

### 3.1. Clinical Relevance

VAP impacts up to 20% of mechanically ventilated patients in the ICU setting, increasing ventilator days, length of stay, cost, and possibly mortality. *Staphylococcus aureus* (*S. aureus*) is a common cause of VAP, with historically reported MRSA rates of up to 60% in the ICU [35]. Rates of methicillin-susceptible *S. aureus* (MSSA) VAP have increased over time, while MRSA VAP has declined from 43.5% in 2012 to 15% in 2021 [36]. Screening for MRSA colonization can be performed using the MRSA surveillance nasal swab, which can detect MRSA via a PCR assay. MRSA nasal PCRs are fast, sensitive, non-invasive, and easy to conduct [37]. A 2014 meta-analysis reported an MRSA colonization rate of 5.8–8.3% in ICU patients. Patients with positive MRSA nasal swabs were found to be more than eight times more likely to have a related infection than non-colonized patients [38].

The 2016 IDSA HAP and VAP guidelines recommend MRSA coverage based on risk factors for multi-drug-resistant pathogens and local susceptibility patterns. Despite the recommendations for initial MRSA coverage, guidelines offer minimal guidance on subsequent de-escalation and streamlining of anti-MRSA antibiotics. The 2016 guidelines do not make recommendations supporting or opposing the use of MRSA nasal surveillance. At the time of publication, there were no prospective trials assessing the role of MRSA screening and empiric antimicrobial choices [2]. Cultures often remain negative, and without screening, healthcare providers have even less to guide de-escalations in therapy [39]. Using only risk factors to guide empiric MRSA coverage can lead to excessive antibiotic therapy [40].

### 3.2. Impact on Antimicrobial Stewardship

#### 3.2.1. MRSA Nares Negative Predictive Values

While the guidelines lack a strong endorsement for MRSA nasal screening utilization in pneumonia, the body of evidence in support of this practice has grown. A meta-analysis conducted in 2018 evaluated MRSA nasal screening and the subsequent development of MRSA pneumonia. Overall, 22 studies with 5163 patients with CAP, HAP, and VAP were included, and the incidence of MRSA pneumonia was 10%. The MRSA screen resulted in a positive predictive value (PPV) of 44.8% and a negative predictive value (NPV) of 96.5%. The NPV among CAP/HAP and VAP patients did not differ significantly, at 98.1% and 94.8%, respectively. There was no impact on clinical outcomes, while simultaneously, the length of MRSA therapy, length of monitoring, and cost were reduced [41]. Another meta-analysis found positive MRSA nares correlate with a relative risk (RR) of 8.33 [95% CI, 3.61–19.20] for MRSA infection when comparing colonized versus non-colonized patients. This meta-analysis found a pooled PPV of 25% and NPV of 97%, correlating to 25% of those with a positive screen developing an MRSA infection vs. 3% of those with a negative screen [38]. Additionally, the long turnaround times for respiratory cultures contribute to overuse of MRSA active therapies and are associated with increased risk of adverse effects, resistance, and cost.

Several studies have examined the utility of screening specifically in the ICU setting and have consistently demonstrated the high NPV of MRSA nasal screening. Research in medical ICUs, trauma ICUs, and mixed ICUs has reported an NPV of >94% in all patients tested [38,40,42,43]. One study evaluated weekly MRSA nasal screens throughout an ICU stay and demonstrated an NPV of 99.4% [42]. While the NPV remains high in the ICU population, the clinical impact of decreased duration of MRSA active therapies is not as frequently seen. This is a missed opportunity for antimicrobial stewardship practices, as many patients should have been de-escalated. Based on these missed opportunities, this is an area for education and a multidisciplinary approach [40].

#### 3.2.2. Timing

A 2018 meta-analysis of 22 studies found that 55% of MRSA screenings were collected on admission, with an additional 31% collected within the first 48 h of admission [41]. A retrospective study of 736 patients split patients into cohorts by time from screen to respiratory culture, ranging from 24 h to more than 14 days. Across all groups, the NPV was 92.9% to 100%, with an overall NPV of 94.9%. These results were similar between ICU and medical/surgical floors [37]. Further supporting this is a study by Bennet et al. looking at time from admission to 30 days afterwards, with an NPV of >98% maintained in groups whether screened within a week or within 30 days [44]. These studies support the use of MRSA PCR results within at least 14 days from the screening.

#### 3.2.3. Implementation Protocols

Some hospitals have implemented pharmacist-driven protocols to facilitate de-escalation based on MRSA nasal screens. A pre–post study described a protocol granting pharmacists authority to order MRSA PCR nasal testing and recommend de-escalation based on results, reducing the duration of anti-MRSA therapy. The average duration of vancomycin therapy decreased from 3.1 days pre-protocol to 1.4 days post (*p* = 0.04) [45]. A similar study compared pharmacy protocols via verbal order (pre) versus collaborative practice agreement (post) for MRSA screening in patients with pneumonia. The screening was higher in the post group: 84% vs. 51% of eligible patients screened, respectively. The duration of therapy was significantly shorter in the post group, as it reached 24 h in the pre-protocol group versus 14.3 h in the post group (*p* < 0.001) [46]. To date, no program has reported a benefit in clinical outcomes, including length of stay, acute kidney injury, readmission rates, total duration of antibiotics, anti-pseudomonal therapy, and mortality. 

Within the ICU setting, screening protocols for MRSA pneumonia have yielded similar findings, with decreased duration of vancomycin therapy (95 vs. 47 h) and decreased vancomycin serum levels with no differences in clinical outcomes [47,48]. One study further evaluated cost savings associated with a pharmacist MRSA screening protocol when vancomycin was ordered for pneumonia. Based on the cost of medication doses, drug levels, PCR testing, and associated personnel, the median cost savings per patient was $40.33, which annualized was over $28,000 [49]. Pharmacist-driven protocols can aid in antimicrobial stewardship efforts, leading to significant cost savings and decreased antimicrobial exposure.

These prior cost analyses included vancomycin therapy and levels [46,47,48,49]. Of note, practitioners should be mindful of changing breakpoints and incidence of vancomycin intermediate *S. aureus* (VISA) and vancomycin-resistant *S. aureus* (VRSA) [50]. A systematic review and meta-analysis evaluated worldwide data and found no evidence of MIC creep over time. However, VISA/VRSA can lead to treatment failure, though the current incidence of VRSA is very rare [51]. MRSA nares can help guide empiric coverage, but definitive therapy should be based on culture data and susceptibilities.

### 3.3. Conclusions

MRSA nares screening has a high NPV for pneumonia in the ICU setting, supporting discontinuation of MRSA antimicrobials when the screen is negative. Studies continue to support quick de-escalation of antimicrobial therapy leading to decreased cost and resources utilized. Implementing a protocol to prompt MRSA nares screening has proven beneficial in regards to duration of antimicrobial therapy and costs to the institution. MRSA PCR nasal swabs play a vital role in both antimicrobial stewardship and management of pneumonia in critically ill patients. 

## 4. Rapid Diagnostics: Rapid Respiratory Panel

### 4.1. Clinical Relevance

The rise in antibiotic resistance continues to make the treatment of HAP a large healthcare burden. Rapid detection and targeted treatment are key to combating this global health issue. Over the last few years, molecular methods have emerged as the diagnostic tools of choice for respiratory pathogens, particularly viruses, owing to their superior sensitivity in the detection of organisms that are difficult to isolate, less viable, or present only in small quantities [52]. New molecular diagnostic tools, such as rapid respiratory panels (RRP), aim at shortening the turnaround time from microbiological sampling to results, which usually takes approximately forty-eight to ninety-six hours, to approximately one to five hours, depending on manufacturer. This accelerated turnaround allows identification of microorganisms and provides potential to target antimicrobial therapy. In addition, RRPs are also less likely to be affected by prior antibiotic administration, unlike standard culture methods, as they do not require viable bacteria [53]. RRPs could possibly provide additional information regarding the presence of antibiotic resistance genes, allowing for earlier de-escalation or escalation of antimicrobial therapy and optimization of antibiotic stewardship. RRPs allow for rapid and comprehensive detection of a wide range of clinically relevant microorganism targets, as well as resistance markers from sputum, endotracheal sputum (ETS), and bronchoalveolar lavage (BAL). Some of the various panels can even provide semi-quantitative results for multiple bacterial pathogens. This could possibly aid in distinguishing clinically relevant pathogens from colonizing bacteria based on the estimates of relative nucleic acid abundance [54].

Clinicians and microbiology laboratory centers now have multiple testing options. Until recently, commercially available nucleic acid amplification tests (NAATs) were limited to viral pathogens plus a few “atypical” bacteria. Deciding which assay or combination of assays to choose and when to use them depends on a variety of factors including the clinical setting, institutional resources, workflow, and cost [55]. Current IDSA guidelines for CAP and HAP/VAP do not address molecular testing for bacterial pathogens other than a recommendation to perform nasal MRSA screening in patients with HAP/VAP [2]. As RRPs are relatively new, their test performance and potential impact on clinical decision making have not yet been fully established. Table 1, Table 2, Table 3 and Table 4 reviews available RRP platforms.

### 4.2. Impact on Antimicrobial Stewardship

#### 4.2.1. Benefits of RRPs

The BioFire^®^ FilmArray^®^ Pneumonia Panel, alongside active antimicrobial stewardship, has been shown to significantly decrease time to appropriate therapy, as well as optimize economic and clinical outcomes [61,62]. A retrospective study showed that an RRP with semi-quantitative results had the potential to lead to a change in empiric antimicrobial therapy in 77% of pneumonia episodes in adult patients with assistance of an antimicrobial stewardship team. The most frequent intervention was de-escalation, which occurred in almost half of the patients [63]. The Flagship II study was a multicenter, randomized controlled trial aimed at determining whether the Unyvero^®^ Hospital Pneumonia multiplex PCR of BAL fluids aided the antimicrobial stewardship team in decision making for patients with pneumonia. The results showed a decrease in inappropriate antibiotic therapy and duration of inappropriate antibiotic therapy from admission in patients with suspected pneumonia, with daily follow-up, to hospital discharge or for a maximum of 30 days. These reductions were without compromise in clinical outcomes (including patient stability, length of stay in hospital, and mortality) [64].

Another area where stewardship has the potential to aid is reducing overall cost associated with various RRPs and determining which patients are the optimal candidates for these tests. This could depend on a variety of factors, such as patients’ comorbidities, immunosuppression, duration of symptoms, severity of disease, and prior antimicrobial use. In addition, while the sensitivity of RRPs decreases the probability of an important pathogen being omitted, amplified detection could overcomplicate interpretation, requiring specialist evaluation. A prospective study, conducted in three ICUs in a French academic hospital (medical ICU, surgical ICU, and cardiothoracic ICU), looked at 95 clinical samples (72 BAL, 23 plugged telescoping catheter) from 85 patients with ventilated HAP or VAP using Unyvero^®^ Hospital Pneumonia (curetis, Holzgerlingen, Germany) multiplex PCR. A multidisciplinary expert panel with microbiological and antimicrobial expertise was utilized to assist in guiding interpretation and integration of RRP results into clinical practice [65].

In a prospective cohort of 100 patients with VAP, the BioFire^®^ FilmArray^®^ (bioMérieux, Salt Lake City, UT, USA) Pneumonia Panel enhanced the positivity rate of conventional diagnostic testing, with increased recognition of co-infections, and reduced time-to-results by more than 48 h [66]. Guillotin et al. showed that as compared to guidance by optimized local protocol or international recommendations, the BioFire^®^ FilmArray^®^ Pneumonia Panel had the potential to reduce the number of days of broad-spectrum antimicrobial therapy (1.7 days [1.4; 2]); *p* < 0.0001) without increasing the risk of treatment failure in critically ill patients with VAP [67]. A similar study evaluated the BioFire^®^ FilmArray^®^ Pneumonia Panel test in a laboratory setting, reporting on 259 adult inpatients submitting BAL specimens for laboratory analysis. It demonstrated a combined 96.2% positive percent agreement (PPA) and 98.1% negative percent agreement (NPA) for the qualitative identification of 15 bacterial targets compared to routine bacterial culture. There was potential for antibiotic adjustment in 70.7% of patients, including discontinuation or de-escalation in 48.2% of patients, resulting in an average savings of 6.2 antibiotic days/patient [68]. Similarly, a prospective article on 95 patients with pneumonia showed that results of the Unyvero^®^ Hospital Pneumonia multiplex PCR could have led to antibiotic changes in 66% of patients. Among the changes, the Unyvero^®^ Hospital Pneumonia multiplex PCR could have led to earlier initiation of an effective antibiotic in 21% of patients, early de-escalation (39%), and optimization (3%). Unyvero^®^ Hospital Pneumonia multiplex PCR also led to two unexpected diagnoses of severe Legionellosis confirmed by culture methods [65]. In a prospective study of 59 samples (BAL, ETS) from 51 ICU patients, the BioFire^®^ FilmArray^®^ Pneumonia Panel results might have led to a de-escalation of initial empirical antibiotics in 16 patients (27.1%), escalation or addition of another effective antibiotic in 9 (15.2%), and no change in 33 (55.9%). The overall performance of the BioFire^®^ FilmArray^®^ Pneumonia Panel was deemed comparable to other RRPs, with overall agreement of greater than 80% for all the available targets tested as well [69].

Proposed benefits of RRPs include high sensitivity, PPV, and NPV. Peiffer-Smadja et al. were able to show that Unyvero^®^ Hospital Pneumonia multiplex PCR was able to identify 104 bacteria vs. 128 by conventional culture within 4.6 hr. When considering the microorganisms isolated at clinical thresholds (104 colony-forming units (CFU)/mL for BAL and 103 CFU/mL for plugged telescoping catheter), 90/112 bacteria were detected by multiplex PCR, which yielded a sensitivity of 80% (95% CI, 71–88%), a specificity of 99% (95% CI, 99–100%), a PPV of 87% (95% CI, 80–93%), and an NPV of 99% (95% CI, 99–99%). The sensitivity was very heterogeneous among bacteria, with overall sensitivity for Gram-negative rods at 90% and for Gram-positive cocci at 62% (*p* = 0.005) [65]. A retrospective study that reviewed 120 respiratory samples (68 sputum, 37 BAL, 15 ETS) utilized the BioFire^®^ FilmArray^®^ Pneumonia Panel and the Unyvero^®^ Hospital Pneumonia multiplex PCR to evaluate the performance of both panels. The BioFire^®^ FilmArray^®^ Pneumonia Panel and the Unyvero^®^ Hospital Pneumonia multiplex PCR identified 82 and 95 pathogens, respectively. The sensitivity/specificity/PPV/NPV, based on culture positivity as a reference, for the Unyvero^®^ Hospital Pneumonia multiplex PCR was 89%/98%/67%/99.6%, and for the BioFire^®^ FilmArray^®^ Pneumonia Panel, it was 95%/98%/62%/99.8%. They showed that both pneumonia panels are sensitive tests in their ability to detect respiratory pathogens with high NPVs. While both panels detect the most common bacterial pathogens, each has unique features that can be useful in different clinical settings [70]). Lee et al., in a prospective study, evaluated the BioFire^®^ FilmArray^®^ Pneumonia Panel compared to traditional culture methods in 51 ICU patients with lower respiratory tract infections. The panel identified a PPV and NPV of 90% [95% CI 73–97.6%] and 97.4% [95% CI 96–98.4%], respectively [69].

In addition, RRPs are also able to detect a variety of resistant organisms, offering the ability to quickly escalate therapy if needed. Peiffer-Smadja et al. showed that the Unyvero^®^ Hospital Pneumonia multiplex PCR was able to detect five *bla*_CTX-M_ among eight (63%) of the ESBL-producing *Enterobacterales* and four carbapenemase genes (*bla*_NDM_ and one *bla*_OXA-23_) out of four carbapenemase-producing *Enterobacterales* (100%). It also detected the only MRSA isolated in conventional culture, but it did not identify any other resistance genes [65]. In regards to the BioFire^®^ FilmArray^®^ Pneumonia Panel, out of four *bla*_CTX-M_ detected, only one case could be verified by the MIC method utilized by Lee et al. In addition, the three carbapenemases observed, except *bla*_VIM_, were consistent with the MIC method and conferred penicillin, cephalosporin, and carbapenem resistance. Two carbapenem-resistant *P. aeruginosa* and one *A. baumannii* were detected by culture but unfortunately not identified by PCR [69].

#### 4.2.2. Limitations

Limitations of these assays include lack of detection for off-target pathogens, a lack of full susceptibility information, cost, and false-positive results. For example, RRP has the potential to detect nucleic acids from dead pathogens not currently causing active infection, which could lead to over-treatment of non-viable microorganisms and potentially deleterious implications for antimicrobial resistance. Implementing protocols and guidelines for effective use of such tests, as well as evaluating novel data, in order to support best practices is vital for antimicrobial stewardship programs to avoid inappropriate use of RRPs. In a prospective study, the Unyvero^®^ Hospital Pneumonia multiplex PCR could have led to three inappropriate antibiotic switches: one inadequacy and two de-optimizations when compared to standard culture methods. More precisely, in one case, the Unyvero^®^ Hospital Pneumonia multiplex PCR identified *Pseudomonas aeruginosa* but missed the presence of an ESBL producing *Enterobacter cloacae* and could have led to a switch from meropenem to ceftazidime. In addition, in two cases, it missed either an *Enterobacter cloacae* or a *Hafnia alvei* and could have led to a switch from cefepime to ceftazidime [65]. Machine failures of over 10% have also been reported in the literature, which is another potential limitation of these various RRPs [71,72]. Substantial discrepancies in the detection of antimicrobial resistance genes were seen by Lee et al., and they recommended complete culture and susceptibility testing be performed in addition to PCR [69]. Consequently, low PPV of the Unyvero^®^ Hospital Pneumonia and BioFire^®^ FilmArray^®^ Pneumonia Panels ranging from 46.9 to 78.6% has been reported in literature, where the comparisons were made assuming that the conventional microbiological techniques are 100% sensitive and specific [73,74]. A laboratory-based evaluation study, which compared the Unyvero^®^ Hospital Pneumonia multiplex PCR and culture results per subject and then explored interpretation of Unyvero^®^ Hospital Pneumonia multiplex PCR false-positive results, which were corroborated by culture from a different sample taken at a later or earlier time point from the same subject, showed reduction in the false-positive results generated from the Unveryo^®^ Hospital Pneumonia multiplex PCR from 29% to 10%, further demonstrating the necessity of continued standard culture techniques in addition to novel diagnostic methods [73]. A prospective study at eight sites in the US evaluating a total of 1682 specimens (846 BAL and 836 sputum) showed >90% sensitivity for most targets on the panel in both BAL and sputum. Some notable exceptions were 75% and 85.7% sensitivity for *Enterobacter aerogenes* in BAL fluid and sputum, respectively. In addition, among the viral targets, adenovirus had low sensitivity in sputum (76.5%), and coronavirus sensitivity was 85.7% and 87.5% in BAL fluid and sputum, respectively. A common extended spectrum β-lactamase recovered in clinical laboratories, CTX-M, had low sensitivities of 85.7% and 80% in both sample types [75]. Faron et al. found similar performance rates for the bacterial targets compared to the standard of care (standard culture methods vs. BioFire^®^ FilmArray^®^ Pneumonia Panel), where the overall PPA was 94.7% (BAL fluid) and 95.8% (sputum). The NPA for the BioFire^®^ FilmArray^®^ Pneumonia Panel was 98.6% (BAL fluid) and 96.5% (sputum). Viral target evaluation was limited, and antibiotic markers were not assessed in the study. Interestingly, targets for *P. aeruginosa* and *Staphylococcus aureus* had lower-than-average PPA of 75% (BAL fluid) and 88.9% (sputum), respectively [76].

Guillotin et al. conducted a cost-effective analysis comparing real-life antimicrobial therapy to simulated antimicrobial therapy using the BioFire^®^ FilmArray^®^ Pneumonia Panel. The cost-effective analysis was carried out over a time horizon that included the resolution of the infectious episode from the hospital perspective. They considered costs related to antimicrobial therapy and BioFire^®^ FilmArray^®^ Pneumonia Panel use, but they did not account for costs from cultures as they were completed in all cases. They estimated that the medical cost was increased by $1349.30 to avoid one day of non-optimized empiric antibiotics. This estimation was based on considering that all patients with HAP would have a BioFire^®^ FilmArray^®^ Pneumonia Panel carried out. This represented 0.6% additional cost to the total hospitalization estimated at $66201.25 per patient, but they only considered the cost impact of the BioFire^®^ FilmArray^®^ Pneumonia Panel on the cost of antimicrobial therapy [67]. Importantly, this cost analysis does not account for any adverse effects due to not treating certain infections as soon as possible (i.e., worsening clinical status/need for more invasive procedures), impact on length of stay or duration of therapy, resistance development, or adverse effects of antimicrobial use, including *Clostridiodes difficile* infection.

### 4.3. Conclusions

Studies have shown a variety of benefits of RRPs, including high sensitivity, specificity, and NPV, when used in conjunction with expert interpretation through utilization of an antimicrobial stewardship team. RRPs could lead to the discontinuation or de-escalation of antibiotic therapy, as well as administration of effective antibiotic therapy to patients. Gullotin et al. believed it was theoretically possible to increase BioFire^®^ FilmArray^®^ Pneumonia Panel efficacy by limiting its indications to use in those patients with MDR bacteria [67]. Lee et al. believed that although the BioFire^®^ FilmArray^®^ Pneumonia Panel may not replace conventional culture and antimicrobial susceptibility testing, especially for bacterial targets, it is still an efficient adjunct to guide clinical decisions and antibiotic treatment in the early stages of pneumonia [69]. Based on this, RRPs have the potential to play an important role in the management of pneumonia in critically ill patients, while assisting in guiding antimicrobial escalation or de-escalation in a timely fashion. Currently, recent CAP, HAP, and VAP guidelines do not address use of RRPs, as they are newer, and their test performance and potential impact on clinical decision making are not yet established [2,3]. In a 2020 IDSA paper, the thought is that RRPs may be most useful in situations where patients have new or worsening lung infiltrates, patients are moderately to severely ill, patients have received empiric antibiotics before obtaining cultures, and/or there is concern for MDR bacteria or polymicrobial infection [55]. It also still remains unclear which patients would benefit the most from RRPs based on previous trials, despite many including a large variety of pneumonia patients. Further studies to evaluate this are needed, which will also aid in determining the cost-effectiveness in various patient populations. In addition, more studies, preferably randomized and multicenter, need to be conducted to determine clinical outcomes, process improvement, which samples are best (i.e., BAL, trach aspirate, sputum), and adverse outcomes.

## 5. Microbiology Reporting Techniques

### 5.1. Clinical Relevance

While advances in microbiological and non-microbiological testing have been, and will continue to be, important in the practice of antimicrobial stewardship, it is clear that communication of “what” and “how” results are provided to practitioners is also important. Clinically relevant information can be reported at nearly every stage of microbiological sample processing.

Blood culture samples are typically collected from patients in both aerobic and anaerobic culture bottles and then incubated in the laboratory for 4 to 7 days. Once growth is detected, samples are removed from incubation, and the first step in organism identification can be performed via a Gram stain. Gram stain status alone has been shown to have a significant effect on antimicrobial selection and should be reported as soon as possible [77]. After Gram staining, samples may continue down several different processing pathways, including molecular testing, mass spectrometry, and phenotypic susceptibility testing, which provide more refined information to guide treatment selection [78].

### 5.2. Impact on Antimicrobial Stewardship

#### 5.2.1. Respiratory Culture Gram-Stain-Directed Therapy

Respiratory cultures follow a slightly different process than blood cultures, but one that yields much the same information. In contrast to blood, the respiratory tract is not a sterile site, but rather is home to a diverse selection of bacteria. This difference requires that respiratory samples undergo an initial screening for quality control to delineate usual respiratory flora from potential pathogens, but also that sample collection be performed judiciously on patients with clinical signs of pulmonary infections [79]. Reporting information, as early as the Gram stain, in respiratory culture processing can have dramatic effects on appropriate antimicrobial use. A randomized clinical trial of 206 ICU patients study compared in-unit Gram-stain-directed antibiotic therapy to traditional guideline-directed therapy for the treatment of VAP. Clinical response rates did not differ between the two groups, at 76.7% vs. 71.8% (*p* < 0.001 [95% CI-0.07–0.17]) for the Gram-stain-directed and guideline-directed therapies, respectively, while the use of anti-pseudomonal and the use of anti-MRSA antimicrobial agents were both reduced by over 30% [80]. Gram staining is not available prior to antimicrobial initiation in most practice settings, however. When staining and organism identification is performed off the unit, in a central microbiology lab, the opportunity for antimicrobial stewardship through reporting Gram stain results remains.

#### 5.2.2. Reporting the Presence of *Staphylococcus aureus* and *Pseudomonas aeruginosa*

In addition to reporting the initial respiratory Gram stain, specifically noting whether *S. aureus* or *Pseudomonas* sp. are present within the sample can be a useful tool in combatting the use of broad-spectrum antimicrobials. A pre–post study of 210 patients found that by changing respiratory sample results from “commensal respiratory flora” to “commensal respiratory flora only: No *S. aureus*/*P. aeruginosa*” for samples without predominant growth of a typical pathogen, there was an 5.5-fold improvement in the odds of de-escalation of anti-MRSA or anti-pseudomonal therapy at the time of result reporting, with odds of 39% in the control group versus 73% in the intervention group (*p* < 0.001 [95% CI 2.8–10.7]) [81]. This important change in culture comments did not change the information that was presented to providers, but rather provided more specific reassurance that common MDR bacteria were not present, which may increase confidence in de-escalation.

#### 5.2.3. Antibiogram Updates

A major turning point in the treatment of infectious diseases is the identification of a causative organism. The traditional antibiogram has long been a helpful tool to guide empiric antimicrobial selection at the stage of organism identification, based on historical institution-specific susceptibility patterns. A novel and potentially lifesaving change to the traditional antibiogram may be the development of a combination antibiogram that details the likelihood of therapy success when two antimicrobials from different classes are used in conjunction, as is recommended for *P. aeruginosa* pneumonia under certain circumstances [3]. Choosing anti-pseudomonal antimicrobials from different classes appears simple at first glance; however, a combination antibiogram may show that the selection of one combination has a higher likelihood of success compared to another. In one such example, if a respiratory isolate is cefepime-resistant, it may be helpful to determine the likelihood that the organism is ciprofloxacin-susceptible. The total cefepime susceptibility in the respiratory isolates is then added to the additional percent that would be covered in the event that cefepime is resistant and ciprofloxacin is susceptible. A retrospective review including data from 304 hospitals found that dual-agent percent susceptibilities for 5157 *P. aeruginosa* isolates from ICU patients depended highly on which two agents were selected. Percent susceptibility for isolates to piperacillin-tazobactam alone was 83.7%, versus combination therapy with a fluoroquinolone improving coverage to 89.8% and combination therapy with an aminoglycoside to 93.4% [82]. An observational study of 17,561 isolates demonstrated another modification to the traditional antibiogram by separating isolates by culture site (e.g., respiratory, blood) and by location of collection (ICU, emergency department, ward), which may provide more specific guidance for therapeutic selection [83]. Combination, location-specific, and syndromic antibiograms all report information already available from current microbiological testing in ways that provide more insight for therapy selection compared to the traditional antibiogram.

#### 5.2.4. Rapid Phenotypic Susceptibility Report

The last and sometimes most important step in microbiological result reporting occurs when antimicrobial susceptibilities are released to prescribers. The time from culture collection to susceptibility reporting is determined by a number of factors, one of which is the time required to incubate and grow cultures. A major limitation is that Clinical and Laboratory Standards Institute (CLSI) guidelines currently recommend an incubation period of 18 to 24 h to prepare an inoculum. Modern laboratory techniques for obtaining antimicrobial susceptibilities typically include the use of broth-microdilution technology via MicroScan Panel (Beckman Coulter, Brea, CA, USA) or Vitek 2 (BioMérieux, Durham, NC, USA). These platforms recommend that cultures first grow on agar plates prior to inoculation in the appropriate technology specific panel. After growth of the organism on an agar plate (typically requiring an 18–24 h incubation), the bacteria can be added to broth microdilution wells. With these modern technologies, we can obtain a full susceptibility report from MicroScan or Vitek 2 within 24 h of inoculation [84]. Therefore, the entire process from plating the sample on the agar plate to growth of the organism and then susceptibility results can take around 34–48 h [85]. Given the delayed time to susceptibility results, alternative methods facilitate obtaining this information more rapidly. Kirby–Bauer disk diffusion is not the most technically advanced laboratory technique; however, it is relatively simple to perform. Additionally, for some newer antimicrobial agents, Kirby–Bauer may be the only way susceptibility tests can be performed [86]. An exploratory microbiology protocol studied agar plates six hours after inoculation. Any colony growth was sub-cultured on a new agar plate, and Kirby–Bauer disks were applied. Authors concluded that an inoculum incubated for only six hours was in agreement with standard 24 h inoculums on 96.7% of the susceptibility tests performed [85]. By performing Kirby–Bauer disk diffusion after six hours of incubation, the time to susceptibility can be decreased from 34–48 h with standard laboratory techniques to 12 h. Standard minimum inhibitory concentrations (MIC) will not be reported by the Kirby–Bauer method; therefore, standard laboratory techniques should still be completed. Antimicrobials reported by the testing laboratory as susceptible are far more likely to be used than if they are not reported at all [87].

#### 5.2.5. Cascade Reporting

Selection of antimicrobial agents to report is an impactful tool to support antimicrobial stewardship efforts if used properly. One strategy for antimicrobial reporting is termed “cascade reporting”, whereby the susceptibilities for agents with broader spectrums of activity are only released if resistance to narrower spectrum alternatives is detected. A pre–post study utilized a cascade reporting system for Gram-negative organisms collected from ICU patients based on ceftriaxone susceptibility. By only reporting cefepime susceptibilities when organisms were found to be ceftriaxone-resistant, the mean number of days of cefepime therapy per encounter was reduced from 1.23 days to 0.81 days (RR: 0.67 [95% CI 0.59–0.75]), and ceftriaxone use increased from 1.49 days to 1.66 days (RR: 1.113 [95% CI 1.009–1.23]). Mortality and readmission rates remained unchanged [88]. This trial is one example of how microbiological reporting can affect prescriber behavior in a targeted manner, at the patient level, but every decision should be weighed against any potential consequences. After implementing a multifactorial selective reporting strategy that considered local susceptibility rates (susceptibilities were not released for drug–organism pairs with a resistance rate >20%), formulary availability, and drug class duplication, a pre–post observational study demonstrated significant changes in antimicrobial resistance patterns. Susceptibility rates for *Klebsiella aerogenes* to amikacin improved from 10% to 100% (*p* < 0.001) and third-generation cephalosporins from 55% to 89% (*p* < 0.001) in a one-year period. It should be noted that the selective pressures of restrictive reporting may have less desired outcomes as well. In the same study, Al-Twfiq et al. found that the removal of amikacin, cefepime, and levofloxacin from *P. aeruginosa* susceptibility reports resulted in a 40% increase in use of piperacillin/tazobactam, correlating with a decrease in susceptibility rates from 99% to 59% (*p* < 0.001) [89]. Restrictive reporting is a powerful tool that can have drastic effects on prescribing practices and potential resistance patterns at institutions. Thoughtful implementation and monitoring for unintended consequences should be considered when utilizing a cascade reporting system for antimicrobial susceptibility reports.

### 5.3. Conclusions

The efficient and effective use of microbiological reporting is an important part of modern antimicrobial stewardship efforts. Early Gram-stain reporting that reinforces the absence of resistant organisms can reduce inappropriate antimicrobial use. Once sensitivity testing is complete, selective reporting of treatment options can be used to enforce formulary restrictions and curb resistance development. Using strategic reporting methods to supplement stewardship efforts can begin as early as the first culture Gram-stain.

## 6. Viral Causes of Pneumonia

### 6.1. Clinical Relevance

Bacterial co-infection in patients with an established viral infection are rare. In two meta-analyses, the incidence of bacterial co-infection in patients with SARS-CoV-2 was low, between 3.5% to 7% [90,91]. In a randomized controlled trial by Wei et al., they enrolled 147 patients. Of these 147 patients, 87 (59.2%) were started on antimicrobial therapy. In this study, no patients were found to have a respiratory bacterial infection, while 7% had a non-respiratory bacterial infection [92]. Overall, this evidence reinforces the need for antimicrobial stewardship in these patients, as many patients that did not need antimicrobials were initially prescribed antibiotics.

### 6.2. Impact on Antimicrobial Stewardship

#### 6.2.1. Procalcitonin in Viral Infections

In the setting of viral infection, interferon (IFN)-γ rises and inhibits the production of PCT [93]. Until recently, little was known about the effect of SARS-CoV-2 on PCT. Early in the COVID-19 pandemic, due to a paucity of data, overuse of antibiotics was rampant, necessitating methods to minimize antimicrobial use. It is now evident that bacterial coinfection rates in COVID-19 are much lower compared to that of influenza. A recent retrospective analysis of 76 patients with COVID-19 found no difference in bacterial coinfections in patients with either PCT > 0.5 ng/mL or <0.5 ng/mL, despite the high-PCT group experiencing longer ICU stays. Notably, these data are limited by the small sample size, antibiotic administration prior to sample obtainment, and exclusion of patients treated with remdesivir or IL-6 receptor antagonists [25]. A retrospective cohort of 95 patients with COVID-19 was categorized by disease severity being mild, moderate, severe, or critical. Mild patients had asymptomatic infection or mild clinical symptoms without any changes in chest imaging. Moderate was defined as patients that had both clinical symptoms of pneumonia and abnormal chest imaging. For patients to be categorized in the severe category, they had to meet one of the following criteria: increased respiratory rate ≥30 breaths/minute, oxygen saturation at rest of ≤93%, or a partial pressure of oxygen/fraction of inspired oxygen (PaO2/FiO2) ratio of ≥300 mmHg. Lastly, critical pneumonia was defined as the rapid progression to meeting one of the following criteria: respiratory failure that required mechanical ventilation, shock, or other organ failure requiring ICU admission. While PCT values correlated well with bacterial coinfection in the moderate disease group, for patients with either severe or critical pneumonia, rates of elevated PCT far exceeded the rates of bacterial coinfection (elevated PCT 50% versus bacterial coinfection 20%; elevated PCT 80% versus bacterial coinfection 50%, respectively) [94]. These findings are consistent with a recent meta-analysis describing a positive correlation between elevated PCT and COVID-19 severity [93]. Given elevated IFN-γ levels have been noted to be associated with increased risk of death in COVID-19 patients (*p* = 0.017), this could explain why PCT is more elevated in patients with severe and critical COVID-19 pneumonia [94,95]. An unexplored limitation of many published studies of PCT and COVID-19 is whether the results will be generalizable to more contemporary variants as the virus continues to mutate.

Data for PCT secretion in patients with influenza pneumonia demonstrate different patterns compared with those seen in COVID-19. In a study of 1608 patients with influenza pneumonia, patients with bacterial co-infection had significantly higher median PCT on admission than patients with no bacterial coinfection, with PCT of 4.35 ng/mL (IQR 0.6–19.9) versus 0.6 ng/mL (IQR 0.2–2.3), respectively, *p* = 0.001. Patients with bacterial coinfection had higher APACHE II scores, more acute renal failure, and more need for renal replacement therapy [96]. In a prospective study of 46 critically ill patients with H1N1 influenza, PCT levels measured within 24 h of ICU admission were elevated in patients with bacterial pneumonia (median = 6.2 ng/mL, *n* = 77) when compared to patients with isolated H1N1 influenza pneumonia (median = 0.56 ng/mL, *n* = 84). This study reported a cutoff of 0.5 ng/mL; PCT had a sensitivity and negative predictive value of 80.5% and 73.2%, respectively [97]. Thus, PCT is reasonable to consider for detection of bacterial pneumonia in patients with concomitant influenza, particularly in patients with community-acquired disease and without immune-compromising disorders.

#### 6.2.2. MRSA Co-Infection with SARS-CoV-2

MRSA does not appear to be more prevalent in the COVID-19 population, as the overall incidence in VAP patients continues to downtrend, including in 2020 and 2021. The incidence of MRSA went from 9.4% in 2012 to 1.3% in 2021 (*p* = 0.001) [36]. Although incidence is low, mortality associated with MRSA superinfection remains high in an already at-risk patient population. An experimental study in Pakistan on 214 patients with MRSA superinfections (≥48 h after admission), all being SARS-CoV-2-positive patients, showed an increase in mortality after 12 to 18 days of hospitalization [98]. A prospective cohort of 148 healthcare-affiliated hospitals noted a correlation of MRSA superinfection with COVID-19 surges, re-enforcing the need to continue infection prevention standards [99]. Infection prevention standards, in addition to antimicrobial stewardship efforts and de-escalation, can ensure the necessary patients are receiving treatment while limiting unnecessary exposure to prevent resistance.

### 6.3. Conclusions

Data are conflicting regarding the impact of PCT in influenza and COVID-19. Although the PCT cutoff of 0.5 ng/mL is commonly used, as a single point, it is not specific enough to determine bacterial co-infection in the setting of either COVID-19 or influenza, as the data demonstrate a high proportion of patients will have a procalcitonin above this level during their course of disease with either virus. Elevated PCT should not be used alone to distinguish the presence of bacterial co-infection for viral pneumonias. PCT levels above 0.5 ng/mL should be interpreted cautiously within the context of other diagnostics, medications, and comorbidities. In regards to MRSA superinfections and COVID-19, there is a low overall incidence of MRSA infections, which were as low as 1.3% in VAP patients in 2021 [36].

## 7. Concluding Remarks

Antimicrobial stewardship is a multifaceted concept that encompasses goals such as averting the spread of antibiotic resistance and preserving efficacy of therapeutic agents without compromising clinical outcomes. Alternatively, better still, improved clinical outcomes may be obtained [100]. Pneumonia is a significant contributor to ICU admission and the use of antimicrobial therapy. Several methods can be employed in a critically ill population to streamline the use of antimicrobials. Utilizing PCT in critically ill patients may yield an increase in antimicrobial use inappropriately. Given the limitations of PCT in the critically ill population, we do not recommend its use to determine the need for initiation of antimicrobials. Utilization of PCT in critically ill patients should be limited to guide duration of antimicrobial therapy and early discontinuation of antibiotics. The efficacy of MRSA PCR-based diagnostic tests in reducing unnecessary antibiotic use through an antimicrobial stewardship team has shown proven benefits throughout the literature, while the promising efficacy of RRPs is still unclear. These tests have the potential to aid providers in decreasing the use of unnecessary and inappropriate antimicrobials, including vancomycin use. As vancomycin use, especially in long durations, is linked to acute kidney injury, avoiding prescriptions and decreasing days of therapy can prevent medication-related toxicities. New technology has also allowed for PCR testing directly of respiratory samples, which can detect our most common causes of HAP and VAP, including MRSA, MSSA, P. aeruginosa, enteric Gram-negative rods such as *Escherichia coli*, and *Acinetobacter baumanii*. If these organisms are not present, de-escalation of broad-spectrum therapy is appropriate. Finally, collaborating with a local microbiology lab to implement antimicrobial stewardship techniques can be an effective means to thwarting broad-spectrum antimicrobial use. Overall, even in the critically ill population, several techniques can be utilized to decrease antimicrobial durations, discontinue unnecessary antimicrobial coverage, and appropriately de-escalate antimicrobial therapy.

## Figures and Tables

**Table 1 antibiotics-12-00295-t001:** Comparison of Various Rapid Respiratory Panels [56].

Panel	Bacterial Target	Viral Targets	Resistance Target	Turnaround Time	Sensitivity and Specificity
BioFire^®^ FilmArray^®^ Pneumonia Panel	**Semi-Qualitative:** *Acinetobactercalcoaceticus-baumannii* complex *Enterobacter cloacae* complex *Escherichia coli* *Haemophilus influenzae* *Klebsiella aerogenes* *Klebsiella oxytoca* *Klebsiella pneumoniae* *Moraxella catarrhalis* *Proteus* spp. *Pseudomonas aeruginosa* *Serratia marcescens* *Staphylococcus aureus* *Streptococcus agalactiae* *Streptococcus pneumoniae* *Streptococcus pyogenes* **Qualitative:** *Chlamydophila pneumoniae* *Legionella pneumophila* *Mycoplasma pneumoniae*	Adenovirus Coronavirus Human metapneumovirus Human rhinovirus/enterovirus Influenza A virus Influenza B virus Parainfluenza virus Respiratory syncytial virus	Carbapenesmases: *bla*_KPC_ *bla*_IMP_ *bla*_NDM_ *bla*_OXA-48_ *bla*_VIM_ Extended-spectrum β-lactamases: *bla*_CTX-M_ Methicillin resistance: *mecA/C* MREJ	~1 h	Sensitivity: 96.2% Specificity: 97.2%

**Table 2 antibiotics-12-00295-t002:** Comparison of Various Rapid Respiratory Panels Continued [57].

Panel	Bacterial Target	Viral Targets	Resistance Target	Turnaround Time	Sensitivity and Specificity
Unyvero^®^ Hospitalized Pneumonia Panel	*Acinetobactercalcoaceticus-baumannii* complex *Chlamydia pneumoniae* *Citrobacter freundii* *Enterobacter cloacae* complex *Escherichia coli* *Haemophilus influenzae* *Klebsiella aerogenes* *Klebsiella oxytoca* *Klebsiella pneumoniae* *Klebsiella variicola* *Legionella pneumophila* *Moraxella catarrhalis* *Morganella morganii* *Mycoplasma pneumoniae* *Pneumocystis jiroveci* *Proteus* spp. *Pseudomonas aeruginosa* *Serratia marcescens* *Staphylococcus aureus* *Stenotrophomonas maltophilia* *Streptococcus pneumoniae*	N/A	Carbapenesmases: *bla*_KPC_ *bla*_IMP_ *bla*_NDM_ Extended-spectrum β-lactamases: *bla*_OXA-23_ *bla*_OXA-23/40_ *bla*_OXA-48_ *bla*_OXA-58_ *bla*_VIM_ *bla*_CTX-M_ Flouroquinolone resistance: *gyrA83* *gyrA87* Macrolide resistance: *ermB* *mecA* *mecC* Penicillin resistance: *bla*_TEM_ *bla*_SHV_ Sulfonamide resistance: *stul 1*	~4.5 h	Sensitivity: 91.4% Specificity: 99.5%

**Table 3 antibiotics-12-00295-t003:** Comparison of Various Rapid Respiratory Panels Continued [58].

Panel	Bacterial Target	Viral Targets	Resistance Target	Turnaround Time	Sensitivity and Specificity
NxTAG^®^ Respiratory Pathogen Panel	*Chlamydophila pneumoniae* *Mycoplasma pneumoniae* *Legionella pneumophila*	Influenza A Influenza A H1 Influenza A H3 Influenza B RSV A RSV B Rhinovirus/Enterovirus Parainfluenza 1 Parainfluenza 2 Parainfluenza 3 Parainfluenza 4 Human Metapneumovirus Adenovirus Coronavirus HKU1 Coronavirus NL63 Coronavirus OC43 Human Bocavirus	N/A	~4 h	Sensitivity: 95.2% Specificity: 99.6%

**Table 4 antibiotics-12-00295-t004:** Comparison of Various Rapid Respiratory Panels Continued [59,60].

Panel	Bacterial Target	Viral Targets	Resistance Target	Turnaround Time	Sensitivity and Specificity
Seeplex^®^ PneumoBacter ACE detection	*Chlamydophila pneumoniae* *Mycoplasma pneumoniae* *Legionella pneumophila* *Bordetella pertussis* *Haemophilus influenzae* *Streptococcus pneumoniae* Internal Control	N/A	N/A	~4 h	Sensitivity: 94.2% Specificity: 96.3%
NeoPlex^®^ RB-8 Detection Kit	*Chlamydophila pneumoniae* *Mycoplasma pneumoniae* *Legionella pneumophila* *Bordetella pertussis* *Haemophilus influenzae* *Streptococcus pneumoniae*	N/A	N/A	~4 h	Sensitivity: 96.2% Specificity: 99.7%

## Data Availability

Not applicable.

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
