# Peer review of "Antimicrobial Stewardship Techniques for Critically Ill Patients with Pneumonia"

_antibiotics, 2023, doi:10.3390/antibiotics12020295_

Round 1

Reviewer 1 Report

Thank you very much for the opportunity to review the manuscript. I read it with interest. In addition to the following comments, I recommend a deep proofreading to ensure the flow and clarity:

1. Introduction
Pneumonia is a common etiology and complication of critical illness, affecting around 27% of all critically ill patients.[1]

1)  This reference is 17 years ago; please, update

2) The first paragraph of introduction: although the authors cited the guidelines, the 4 types and the last sentence need further citations.

3) This data supports guideline recommendations to provide initial antimicrobial coverage for MRSA and PSAR in patients with HAP and VAP.

Which guideline? reference 2? please, cite again.

4) The Centers for Disease Control and Prevention (CDC) describes antimicrobial resistance as an “urgent global public health threat, killing at least 1.27 million people worldwide and associated with nearly 5 million deaths in 2019”.

Please, cite the reference or the web page

5) The authors are encouraged to clarify the flow of information in the Procalcitonin section

6) Procalcitonin conclusion. Please, cite again the refence at the end: The use of PCT-guided antibiotic cessation protocols was shown by Lam et al to decrease mortality.

7) Move the following to a new row:

3. Methicil-lin-resistant Staphylococcus aureus nasal polymerase chain reaction utilization

8) Page 4: VAP mention in full at the begining

Please, modify the following:

3.2 Impact on Antimicrobial Stewardship3.2.1 MRSA Nares Negative Predictive Valus

9)As the authors included MRSA and vancomycin, adding more data about VISA, VRSA, and MIC creep and relation with stewardship is encouraged.

10) In the concluding remarks, the following is inappropriate: (Tamma et al ref), please, change

Author Response

Thank you very much for taking the time to review our article and provide meaningful suggestions. Below we addressed each of the comments individually.

Reviewer's Comments:

Introduction
Pneumonia is a common etiology and complication of critical illness, affecting around 27% of all critically ill patients.[1]

1)  This reference is 17 years ago; please, update

Author's Response: 

Added in: “according to an article published in 2006”. Reviewed newer references, however incidence was specific to mortality in VAP or HAP patients. This was the best estimate for incidence of pneumonia in critically ill patients which was the intent of the author. We are attempting to convey that this is a high burden disease, not necessarily point out the high mortality rate.  

Reviewer's Comments:

2) The first paragraph of introduction: although the authors cited the guidelines, the 4 types and the last sentence need further citations.

Author's Response:

The 4 types of pneumonia are referenced within the 2 guidelines. The citation has been moved to the end of the last sentence.  

Reviewer's Comments:

3) This data supports guideline recommendations to provide initial antimicrobial coverage for MRSA and PSAR in patients with HAP and VAP.

Which guideline? reference 2? please, cite again.

Author's Response:

Addressed. Please see track changes in the attached document.

Reviewer's Comment:

4) The Centers for Disease Control and Prevention (CDC) describes antimicrobial resistance as an “urgent global public health threat, killing at least 1.27 million people worldwide and associated with nearly 5 million deaths in 2019”.

Please, cite the reference or the web page

Author's Response:

The reference is number 6 that is sited after the next sentence. The authors were attempting to not cite the same reference for multiple sentences in a row, therefore this edit has not been addressed in the revisions.  

Reviewer's Comments: 

5) The authors are encouraged to clarify the flow of information in the Procalcitonin section

Author's Response:

Thank you for the suggestion. Please advise if further changes are needed to improve flow. The following changes were made directly to the manuscript: 

1) The final sentence in the section 2.1 Clinical Relevance was modified to improve flow. It now reads as follows: “Despite recommendations against routine use as a diagnostic tool, procalcitonin may be considered in certain clinical scenarios to support objective antibiotic de-escalation as described further in the sections that follow.” 

2) The final statement in the section 2.3 Conclusion was removed and the preceding two sentences were modified to the following: “Overall, PCT has limited utility in antimicrobial initiation and clinicians should be cautious with interpretation of results to guide diagnosis and initial antibiotic prescribing. However, PCT may provide a benefit for antimicrobial stewardship in the ICU as a de-escalation tool.” 

Reviewer's Comment:

6) Procalcitonin conclusion. Please, cite again the refence at the end: The use of PCT-guided antibiotic cessation protocols was shown by Lam et al to decrease mortality.

Author's Response:

Thank you for pointing this out. The final statement requiring a reference was removed from the manuscript to improve flow and concluding comments. 

Reviewer's Comments:

7) Move the following to a new row:

3. Methicil-lin-resistant Staphylococcus aureus nasal polymerase chain reaction utilization

Author's Response:

Addressed in the track changes of the attached document. 

Reviewer's Comments:

8) Page 4: VAP mention in full at the begining

Author's Response:

Thank you for pointing this out. It has been addressed. Please see track changes in the attached document.

Reviewer's Comments:

Please, modify the following:

3.2 Impact on Antimicrobial Stewardship3.2.1 MRSA Nares Negative Predictive Valus

Author's Response:

Thank you for pointing this out. It has been addressed. Please see track changes in the attached document.

Reviewer's Comments:

9)As the authors included MRSA and vancomycin, adding more data about VISA, VRSA, and MIC creep and relation with stewardship is encouraged.

Author's Response: 

Added in a paragraph under 3.2.3 (see track changes). It was a very brief overview as more information is out of scope for the topic of MRSA nares utilization for pneumonia stewardship. 

Reviewer's Comments:

10) In the concluding remarks, the following is inappropriate: (Tamma et al ref), please, change

Author's Response:

Thank you for pointing this out. It has been addressed. Please see track changes in the attached document.

Reviewer 2 Report

Thank you for the opportunity to review this outstanding manuscript. It was a pleasure to read.

The manuscript addresses the diagnostic and therapeutic management utility of procalcitonin, MRSA nares screening, rapid respiratory panels in critically ill patients with suspected or confirmed pneumonia. These novel techniques are discussed in the context of antibiotic selection, duration of therapy, and de-escalation, along with antimicrobial stewardship. Much of the information presented concerns trials conducted after publication of the 2016 IDSA guidelines for HAP and VAP. The review is thorough, applicable to current practice, and based on current and appropriate references. The authors' conclusions are appropriate based on the literature included in the review.

Author Response

Thank you very much for your insightful and positive review of our article. We appreciate your assistance with ensuring this is a high quality article.

Reviewer 3 Report

The manuscript is well-written and organized.

The authors should revise the manuscript as throughout it “Unyvero” is sometimes incorrectly mentioned as “Unveryo”.

Moreover, some sentences seem unfinished or not correctly formed. For instance, l. 26-29 and l. 71-74 from section 4.2.1. The latter could be improved if merged with the following sentence.

In my understanding, this review should also include at least a mention of the impact of NGS on this field, with metagenomic sequencing or whole genome sequencing. An example of the usage of the latter can be found in Forde et al. 2022 (https://doi.org/10.1093/cid/ciac726).

Minor comments:

- Remove the numbering of the keywords.

- Formatting should be corrected when introducing section 3 as it is currently on the same line as the last sentence of the previous section. (same thing for 3.2 and 3.2.1)

- In section 3.2.1, is “valus” the intended word, or could it be a typo?

- In section 7, the “(Tamma et al ref)” should be included/replaced (l.417)

Author Response

Reviewer's Comments:

The authors should revise the manuscript as throughout it “Unyvero” is sometimes incorrectly mentioned as “Unveryo”.

Author's response:

Thank you for pointing this out. This has been addressed  and can be viewed within the track changes of the attached document. 

Reviewer's Comments: 

Moreover, some sentences seem unfinished or not correctly formed. For instance, l. 26-29 and l. 71-74 from section 4.2.1. The latter could be improved if merged with the following sentence.

In my understanding, this review should also include at least a mention of the impact of NGS on this field, with metagenomic sequencing or whole genome sequencing. An example of the usage of the latter can be found in Forde et al. 2022 (https://doi.org/10.1093/cid/ciac726).

Author's Response: 

1) Incomplete sentences addressed 

2) Regarding NSG: Although NGS is of importance, it is generally blood samples and not standard of care.  This review is specifically for PNA and PNA PCRs that are obtained through respiratory cultures. Given this, we felt that it was out of scope for our topic. 

Reviewer's Comments:

Remove the numbering of the keywords.

Author's Response:

Thank you for pointing this out. This has been addressed  and can be viewed within the track changes of the attached document. 

Reviewer's Comments:

 Formatting should be corrected when introducing section 3 as it is currently on the same line as the last sentence of the previous section. (same thing for 3.2 and 3.2.1)

Author's Response:

Thank you for pointing this out. This has been addressed  and can be viewed within the track changes of the attached document. 

Reviewer's Comments:

In section 3.2.1, is “valus” the intended word, or could it be a typo?

Author's Response:

Thank you for pointing this out. This has been addressed  and can be viewed within the track changes of the attached document. 

Reviewer's Comments:

In section 7, the “(Tamma et al ref)” should be included/replaced (l.417

Author's Response:

Thank you for pointing this out. This has been addressed and can be viewed within the track changes of the attached document. 

Round 2

Reviewer 1 Report

The manuscript has been improved